# Very High Resolution Automotive SAR Imaging from Burst Data

**Mattia Giovanni Polisano** [1,*], **Marco Manzoni** [1], **Stefano Tebaldini** [1], **Andrea Monti-Guarnieri** [1], **Claudio Maria Prati** [1] **and Ivan Russo** [2]

[1] Department of Electronics, Information and Bioengineering (DEIB), Politecnico di Milano, 20133 Milan, Italy
[2] Huawei Technologies Italia S.r.l., 20129 Segrate, Italy
[*] Correspondence: mattiagiovanni.polisano@mail.polimi.it

**Abstract:** This paper proposes a method for efficient and accurate removal of grating lobes in automotive Synthetic Aperture Radar (SAR) images. Grating lobes can indeed be mistaken as real targets, inducing in this way false alarms in the target detection procedure. Grating lobes are present whenever SAR focusing is performed using data acquired on a non-continuous basis. This kind of acquisition is typical in the automotive scenario, where regulations do not allow for a continuous operation of the radar. Radar pulses are thus transmitted and received in bursts, leading to a spectrum of the signal containing gaps. We start by deriving a suitable reference frame in which SAR images are focused. It will be shown that working in this coordinate system is particularly convenient since it allows for a signal spectrum that is space-invariant and with spectral gaps described by a simple one-dimensional function. After an inter-burst calibration step, we exploit these spectral characteristics of the signal by implementing a compressive sensing algorithm aimed at removing grating lobes. The proposed approach is validated using real data acquired by an eight-channel automotive radar operating in burst mode at 77 GHz. Results demonstrate the practical possibility to process a synthetic aperture length as long as up to 2 m reaching in this way extremely fine angular resolutions.

**Keywords:** radar; MIMO; SAR; automotive; grating lobes suppression; compressive sensing; automotive radar; CLEAN





## 1. Introduction

Advanced Driver Assistant Systems (ADAS) are increasingly being considered in the context of the research towards autonomous driving vehicles. By and large, ADAS require abundant and reliable data acquired by various sensors, such as cameras, LiDARs, and radars. Automotive radars, in particular, have proven to be excellent tools for sensing the surrounding environment thanks to the day and night imaging capability, and the ability to measure distances and velocities with great accuracy. Although radars are used for ADAS systems, the interest in the development of this technology in the automotive scenario has been growing widely in the last decades [1–5].

Current automotive radars generally provide coarser resolution than optical and LiDAR sensors, which is perceived as an intrinsic limitation of such technology. Indeed, vehicle-based radar imaging is typically implemented using Multiple-Input Multiple-Output (MIMO) devices, for which angular resolution is bounded by the number of Tx/Rx channels forming the MIMO array. For this reason, achieving fine angular resolution requires the implementation of more complex, large, and expensive devices. Nowadays, radars used for automotive purposes operate in the W-band [6] (above 76 GHz) and are mostly used for crash avoidance and object detection at short, medium or long range depending on their wavelength.

To overcome such limitations, recent researches have focused on the concept of synthetic aperture radar (SAR), where an arbitrarily long array is formed by exploiting the

natural motion of the vehicle [1–4]. Still, a necessary condition for any conventional SAR imaging algorithm is that the radar pulses are acquired continuously, meaning that the device is constantly operating. Unluckily, this is not the case in the current automotive scenarios: due to radio-frequency regulations and hardware requirements, radar devices are typically designed to operate on a non-continuous basis by sending bursts of few hundreds of pulses alternated with periods of silence. This acquisition mode creates gaps in the spatial spectrum of SAR images, which manifest themselves through the appearance of grating lobes in SAR images, leading to false target detections. This problem is typical of SAR imaging and it has been widely covered in the literature [7,8].

The aim of this paper is to design, implement and test a routine able to remove grating lobes from SAR images generated by processing data acquired in bursts. The goal is achieved with several steps. First of all, a space-invariant reference system is derived in which the spectrum of the data depends uniquely on the position of the sensor and not on the position of the target. Unlike more common reference systems such as the Cartesian or the polar ones, this reference frame allows us to see the gaps in the spectrum of the data. The high resolution SAR image (still showing grating lobes) will be the coherent sum of the bursts. Each burst, however, must be focused properly using autofocusing techniques [3] in order to correct for possible trajectory errors. Then, each burst must be phase calibrated [9,10] in order to allow for a fully coherent summation. Both steps are described carefully in this paper. Finally, the removal of grating lobes is achieved with the CLEAN algorithm, developed in [7] for speckle and grating lobes suppression in astronomy and it is widely used also in telecommunications [11–14].

This paper is organized as follows: Section 2 covers the geometry and the signal model, and the basic notions of automotive SAR are dealt. Section 3 regards the wavenumber domain in SAR imaging, where the theory regarding the spatial spectral domain characteristics of a burst acquired signal are covered. In Section 4, the new mathematical framework which is the corner stone of this work and that allows a space-invariant spatial spectrum descriptor is developed, also the autofocus, phase calibration technique and the grating lobes suppression algorithm are described in this section. Section 5 shows the real data results used to validate this work and finally in Section 6, the conclusions are drawn.

## 2. Geometry, Signal Model and SAR Processing

In this paper we will consider a MIMO radar having $N_{TX}$ transmitting antennas and $N_{RX}$ receiving ones, forming an equivalent monostatic virtual array of $N_{ch} = N_{TX} \times N_{RX}$ elements. The elements and their spatial locations are represented in Figure 1.

The radar is mounted with a arbitrary installation angle on the car. A forward looking geometry is indicated for target detection and navigation assistance; a side looking geometry is, instead, more indicated for parking detection, while a squinted geometry can be useful for collision avoidance. From now on, we will consider a forward looking geometry with the virtual elements displaced along the $y$ direction.

It is well known that a system such as the one just described will provide a slant range resolution equal to [15]:

$$\rho_r = \frac{c}{2B} \tag{1}$$

where $c$ is the speed of light and $B$ is the bandwidth of the radar pulse. The angular resolution, instead, is related to the length of the virtual array as:

$$\rho_\phi^M = \frac{\lambda}{2N_{ch}\Delta x \cos\phi} = \frac{\lambda}{2L_M \cos\phi} \tag{2}$$

where $\Delta x$ is the distance between elements of the virtual array, $L_M$ is the total length of the virtual array, $\phi$ is the off boresight angle as depicted in Figure 1.

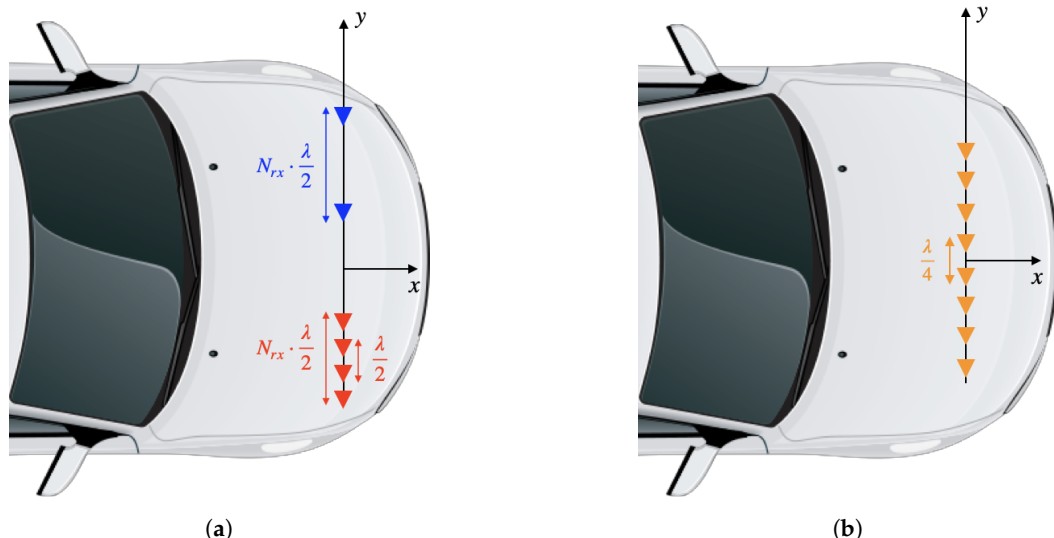

<div align="center">(<b>a</b>)             (<b>b</b>)</div>

**Figure 1.** (**a**): Real array configuration for an 8-channel automotive radar made up by two transmitting antennas (in blue) and four receiving antennas (in red) in forward-looking configuration. (**b**): Virtual array for an 8-channel automotive radar spaced by $\lambda/4$ in forward-looking configuration.

In order to avoid ambiguities, the distance between the virtual elements should be equal to $\lambda/4$ [16]. With this spacing, if we take as an example a radar with eight virtual channels working at 77 GHz, the finest achievable resolution is roughly fifteen deg.

One way to improve angular resolution is by exploiting the well known SAR concept. While the MIMO array is transmitting pulses, the vehicle where it is mounted moves, generating the so-called synthetic aperture. While for the MIMO radar the resolution was bounded by the length of the virtual array, now the resolution is bounded just by the length of the aperture as

$$\rho_\phi^{SAR} = \frac{\lambda}{2L_{A_s}sin(\phi)} \tag{3}$$

where $L_{A_s}$ is the length of the synthetic aperture. Notice that now at the denominator there is a *sin* function. While the MIMO array provides the best resolution at its boresight (in front of the car), the SAR image has its best resolution in the direction orthogonal to the motion (i.e., at the left or right of the car).

It is now useful to go deeper into the signal model and in SAR processing (also called SAR focusing). Without loss of generality, we assume that the vehicle travels in a rectilinear trajectory along *x*. We remark that this is not a necessary condition for SAR imaging to work, but it simplifies the following derivations. The car travels with a velocity equal to $v_x$ and the MIMO radar transmits a linearly frequency modulated chirp every Pulse Repetition Interval (PRI) second. Given the velocity, the radar system adapts its PRI in order to mantain a spatial sempling distance of [16]

$$PRI \times v_x = \frac{\lambda}{4} \times N_{ch} \tag{4}$$

where $N_ch$ is the number of MIMO channels. A visual representation is given in Figure 2.

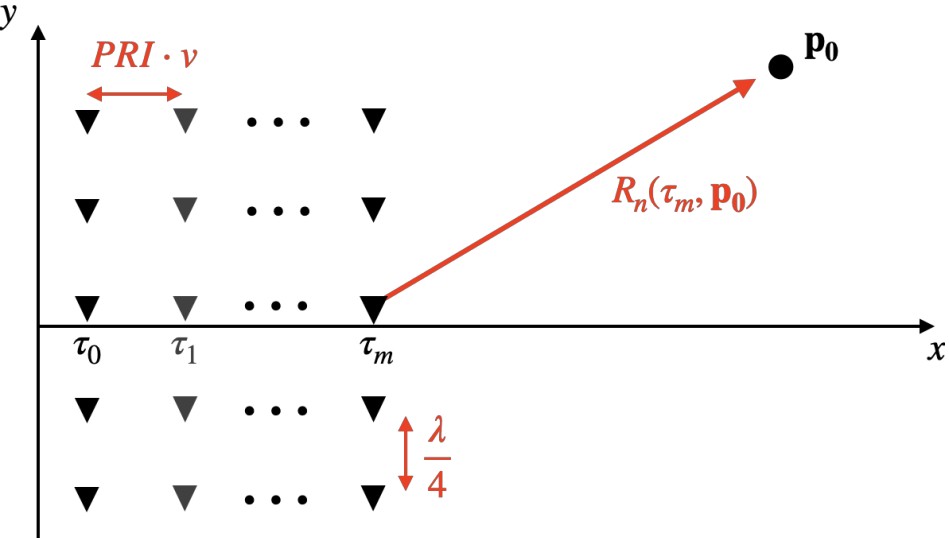

**Figure 2.** Uniform Linear array with spacing distance $d = \lambda/4$ in forward SAR looking configuration, with a reference point target $\mathbf{p_0}$ moving at a speed $v$ with a Pulse Repetition Interval *PRI*.

After the reception of the echo, the signal is range compressed. The signal model of the range compressed signal is [17]

$$s_n(r, \tau; \mathbf{p_0}) = \text{sinc}\left[\frac{r - R_n(\tau; \mathbf{p_0})}{\rho_r}\right] \times \exp\left\{-j\frac{4\pi}{\lambda}R_n(\tau; \mathbf{p_0})\right\} \tag{5}$$

where

- $r$ is the slant range;
- $\tau$ is the slow time sampled every PRI seconds;
- $n$ is the index of the virtual element;
- $\mathbf{p_0} = [x_0, y_0]^{\text{T}}$ is the geometrical position of the target;
- $R_n(\tau; \mathbf{p_0})$ is the distance from the $n^{th}$ virtual element to the target at time $\tau$.

Notice that in Equation (5), we have neglected an amplitude factor and all the bistatic effects (i.e., we are considering an ideal monostatic equivalent of the MIMO TX/RX scheme).

We can now exploit a set of samples to form a SAR image by using a simple but straightforward time domain back projection (TDBP) algorithm [16]:

$$I(\mathbf{p}) = \sum_{\tau}\sum_{n=1}^{N_{ch}} s_n[R_n(\tau, \mathbf{p}), \tau; \mathbf{p_0}] \times \exp\left\{+j\frac{4\pi}{\lambda}R_n(\tau, \mathbf{p})\right\} \tag{6}$$

where $I(\mathbf{p})$ is the SAR image at pixel $\mathbf{p} = [x, y]^{\text{T}}$, the expression $s_n[R_n(\tau, \mathbf{p}), \tau; \mathbf{p_0}]$ indicates the range compressed signal evaluated at a range $R_n(\tau, \mathbf{p})$, which is the distance between the $n^{th}$ virtual phase center and the generic pixel $\mathbf{p}$ at time $\tau$.

We highlight that Equation (6) can be split into two separate summations. First, we form a set of low-resolution snapshots by coherently summing over the channels:

$$I_M(\tau, \mathbf{p}) = \sum_{n=1}^{N_{ch}} s_n[R_n(\tau, \mathbf{p}), \tau; \mathbf{p_0}] \times \exp\left\{+j\frac{4\pi}{\lambda}R_n(\tau, \mathbf{p})\right\} \tag{7}$$

where $I_M(\tau, \mathbf{p})$ is the low-resolution image generated by the MIMO radar at time $\tau$. We now sum over all the slow times, obtaining the final SAR image:

$$I(\mathbf{p}) = \sum_{\tau} I_M(\tau, \mathbf{p}) \tag{8}$$

The two-step summation opens the possibility for advanced and efficient focusing schemes [4] and autofocusing algorithms to refine the vehicle trajectory [3,18].

## 3. Wavenumbers in SAR Images

In this section, we will recall the concept of *wavenumber* which is widely used in geophysical, radar and SAR imaging. Any SAR experiment can be described in two conjugate domains: the direct domain referred to usually as time or space domain, and the frequency or wavenumber domain. This representation comes useful since any SAR experiment can be considered as the enlightenment of the spatial frequency components of a target, as extensively explained in [19].

If we focus for a moment on a simple Cartesian 3D reference system, the transformation from the spatial domain to the wavenumber domain can be performed by a simple 3D Fourier transform, that is by projecting the spatial signal on the basis function:

$$e^{j\mathbf{k}\cdot\mathbf{r}} = e^{j(k_x x + k_y y + k_z z)} = e^{j\psi} \tag{9}$$

where $\mathbf{k} = [k_x, k_y, k_z]^T$ is the so-called *wavevector* and its components are called *wavenumbers*. The wavenumbers can be computed by simply taking the partial derivative of the phase $\psi$ w.r.t. of each spatial variable. Taking advantage of the gradient notation:

$$\mathbf{k} = \nabla\psi \tag{10}$$

where $\nabla$ is the gradient operator and $\psi$ is the phase in Equation (9). The transmitted signal can be represented in the spatial domain (in the far field approximation) as a plane wave illuminating the target. The direction of propagation of such a wave is defined by the wavevector $\mathbf{k}$ and it is parallel to the line joining the sensor and the target. It follows that a movement of the sensor corresponds to a change in the illuminated wavenumber, or, in other words, to an expansion of the bandwidth of the signal in the wavenumber domain.

To clarify the need of a custom reference frame in place of a Cartesian or polar one, we must highlight a few characteristics of the signal in the wavenumber domain, such as the dependence of the illuminated wavenumber by both the position of the sensor and the target in the scene. This issue is discussed more in detail in the next section.

## 4. Imaging with Bursted Data

This section encloses the heart of this work. First, a description of the burst acquisition mode is provided as well as the spectral coverage of the most common reference systems and the derivation of a proper reference system where the spectral space invariance property holds. Second, the procedure for burst phase calibration is provided and finally, the CLEAN algorithm is described. A visual representation of our workflow is given in Figure 3.

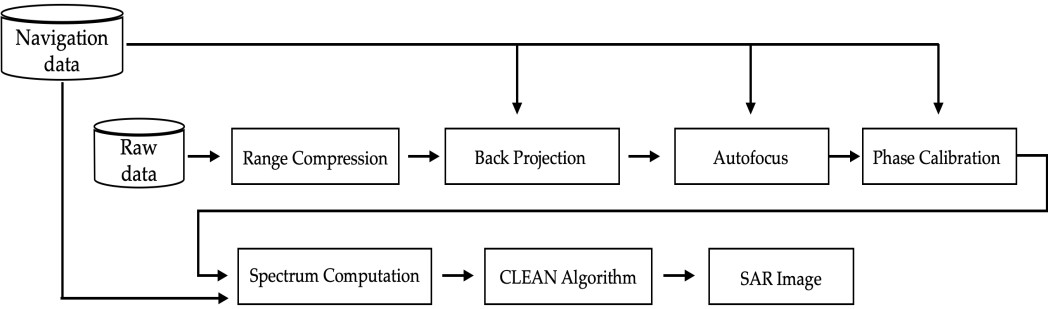

**Figure 3.** Grating lobes suppression workflow block diagram.

### *4.1. Burst Data Acquisitions*

As described in Section 3, the reciprocal position of radar and target determines the illuminated wavenumbers and thus the SAR angular resolution. Usually, the processed synthetic aperture is uniformly sampled; therefore, the coverage in the wavenumber domain is uniform (without "holes").

In some cases, and in particular in the automotive radar, SAR acquisitions may be performed on a non-continuous basis. The reason behind this lies on the regulations establishing the coexistence between different telecommunication devices [20] and for electronic battery power saving. In practice, this acquisition mode, called *burst mode*, consists in turning on and off, periodically, the radar along the traveled trajectory. Figure 4 explains the concept clearly.

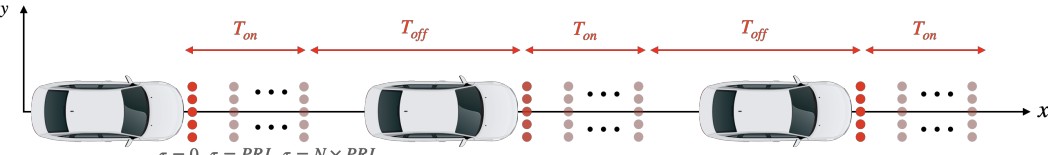

**Figure 4.** Burst acquisition mode representation: while the vehicle is moving the radar works for a period of time $T_{on} = N \times PRI$, where $N$ is the number of slow time samples for each burst, and it is silent for a period of time $T_{off}$.

The result of this non-continuous acquisition is that, in the spatial frequency domain, some wavenumbers are not covered, generating some gaps in the spectrum. In the focused image, these gaps manifest themselves as artifacts commonly known as grating lobes. We recall once again that the positions of the covered areas (from now on called *tiles*) and the positions of the gaps in a Cartesian reference frame are a function of both the radar position and the target position. For an artificial problem composed by a single target, the gaps will be visible in the spectrum, but this is not the case if the scene is composed by a variety of different targets distributed in any position of the field of view.

The need arises for suitable data representation that allows a common spectral description for all the targets in the image in order to allow a standard signal processing. In the following section, a mathematical framework will be developed in order to overcome this problem.

### *4.2. 2D Tessellation*

In this subsection, we are going to analyze the spectral behavior of the most common reference systems: we start from the simplest one which is the Cartesian; then, we will proceed with a frequency description of the signal in the polar coordinates and finally on a custom (and appropriate for our needs) reference frame.

#### 4.2.1. Cartesian Reference System

Given a 2D Cartesian reference system as in Figure 5 and a target located in **p**, the range equation for a given time instant $\tau$ is

$$R(\tau; \mathbf{p}) = \sqrt{(x(\tau) - x_p)^2 + (y(\tau) - y_p)^2} \tag{11}$$

where

- $x(\tau)$ is the position of the radar along the $x$ axis at time $\tau$;
- $y(\tau)$ is the position of the radar along the $y$ axis at time $\tau$;
- $(x_p, y_p)$ are the coordinates of the target.

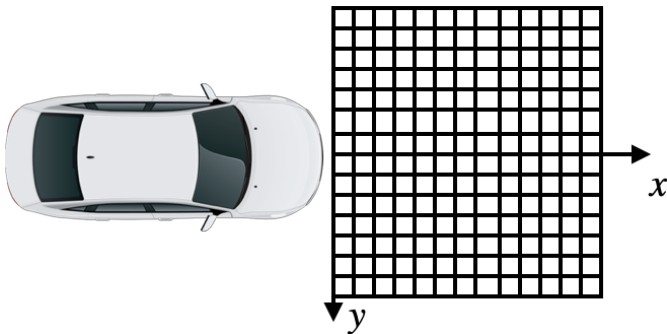

**Figure 5.** Back projection grid representation in a Cartesian reference system for a forward looking Automotive SAR.

The spectral contributes $(k_x, k_y)$ can be computed as the partial derivative of the range equation w.r.t. for each variable:

$$k_x = \frac{4\pi}{\lambda} \frac{\partial R(\tau; \mathbf{p})}{\partial x} = \frac{4\pi}{\lambda} \frac{x(\tau) - x_p}{R(\tau; \mathbf{p})} = \frac{4\pi}{\lambda} \cos \theta(\tau) \tag{12}$$

$$k_y = \frac{4\pi}{\lambda} \frac{\partial R(\tau; \mathbf{p})}{\partial y} = \frac{4\pi}{\lambda} \frac{y(\tau) - y_p}{R(\tau; \mathbf{p})} = \frac{4\pi}{\lambda} \sin \theta(\tau) \tag{13}$$

where $\theta(\tau)$ is the look angle from the boresight of the radar to the target. The above equations are obviously space variant, i.e., they depend on both the position of the target $\mathbf{p}$ and the position of the radar.

### 4.2.2. The Polar $(r, \theta)$ Reference System

Another common reference system is the polar one., depicted in Figure 6. It is possible to fix a relationship between the polar and the Cartesian system as

$$\begin{cases} x = r \cos \theta \\ y = r \sin \theta \end{cases} \tag{14}$$

where $r$ is the range and $\theta$ the angle.

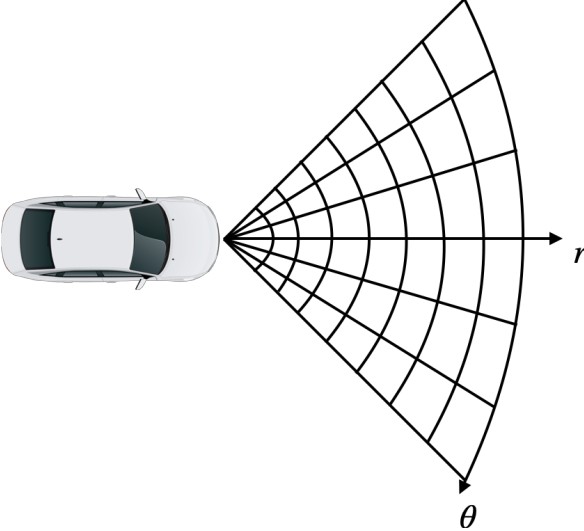

**Figure 6.** Back projection grid representation in a Polar reference system for a forward looking Automotive SAR.

It is rather easy, thanks to the derivative chain rule, to compute the spectral contributes under this reference system $(k_r, k_\theta)$, as

$$
\begin{cases}
k_r = \dfrac{4\pi}{\lambda}\left(\dfrac{\partial R(\tau; \mathbf{p})}{\partial x}\dfrac{\partial x}{\partial r} + \dfrac{\partial R(\tau; \mathbf{p})}{\partial y}\dfrac{\partial y}{\partial r}\right) = \dfrac{4\pi}{\lambda}\left[\cos\big(\theta(\tau) - \theta_p\big)\right] \\[2mm]
k_\theta = \dfrac{4\pi}{\lambda}\left(\dfrac{\partial R(\tau; \mathbf{p})}{\partial x}\dfrac{\partial x}{\partial \theta} + \dfrac{\partial R(\tau; \mathbf{p})}{\partial y}\dfrac{\partial y}{\partial \theta}\right) = \dfrac{4\pi}{\lambda}r\left[\sin\big(\theta(\tau) - \theta_p\big)\right]
\end{cases}
\tag{15}
$$

where $\theta(\tau)$ is the angle between the sensor and the target and $\theta_p$ is the angular position of the target in the polar reference frame. If we place the origin of the polar grid at the center of the aperture and we assume a conveniently small aperture where $\theta(\tau) \approx \theta_p$, we can expand Equation (15) in Taylor series around the point $\theta(\tau) - \theta_p = 0$, obtaining

$$
k_r(\tau) = \frac{4\pi}{\lambda}\left[1 - \frac{1}{2}d\theta(\tau)^2\right]
\tag{16}
$$

which can be considered almost constant for short enough apertures. For the angular wavenumber, instead, we have:

$$
k_\theta(\tau) = \frac{4\pi}{\lambda}\left[r \cdot d\theta(\tau)\right]
\tag{17}
$$

which, instead, is not constant. Since the dependency on the position of the radar is not perfectly highlighted, let us use the relation (14) to explicit the differential $d\theta$ as a function of $(x, y)$:

$$
d\theta = \frac{1}{r}\left[-dx \cdot sin(\theta) + dy \cdot cos(\theta)\right]
\tag{18}
$$

in order to refer Equation (17) explicitly to the radar position, it is possible to look at the differentials $dx$ and $dy$ as the displacement of the sensor position along the trajectory, namely

$$
k_\theta(\tau) = \frac{4\pi}{\lambda}\left[-\Delta x(\tau) \cdot sin(\theta_p) + \Delta y(\tau) \cdot cos(\theta_p)\right]
\tag{19}
$$

which depends on both the sensor position and the position of the target. Again, the polar reference frame is space variant and therefore not suitable for our purpose.

### 4.2.3. The (r,e) Reference System

In this section, we develop a reference system such that the illuminated wavenumbers are uniquely dependent on the position of the sensor and not on the position of the targets in the scene. We call this system $(r, e)$, since one variables remains the range (which is space invariant) and the other is instead a transformation of the angle $\theta$, such that the resulting wavenumber is independent upon the target position:

$$
\begin{aligned}
k_e(\tau) &= k_\theta(\tau)\frac{d\theta}{de} \\[2mm]
&= \frac{4\pi}{\lambda}\left[-\Delta x(\tau) \cdot sin(\theta) + \Delta y(\tau) \cdot cos(\theta)\right] \cdot \frac{d\theta}{de}
\end{aligned}
\tag{20}
$$

There is no particular solution $d\theta/de$ allowing a total independence of $k_e(\tau)$ from the target position. However, if we assume that the target trajectory is along only one direction, let us say $x$, we can discard the other term by forcing $\Delta y(\tau) = 0$. Therefore (20) becomes

$$
k_e(\tau) \simeq \frac{4\pi}{\lambda}\left(-\Delta x(\tau) \cdot sin(\theta)\right) \cdot \frac{d\theta}{de}
\tag{21}
$$

It comes straightforward that in order to have space invariance the ratio $d\theta/de$ must be equal to

$$\frac{d\theta}{de} = \frac{1}{sin(\theta)} \tag{22}$$

and therefore:

$$\frac{de}{d\theta} = sin(\theta) \tag{23}$$

From which we can derive the change of variable needed:

$$e(\theta) = \int_{-\frac{\pi}{2}}^{\theta} sin(\theta') \cdot d\theta' = cos(\theta) \tag{24}$$

The reference system built upon this assumption is ambiguous at bore-sight, since the function $cos(\theta)$ is even. One possible solution is to process separately two sides of the scene, left and right. However, this solution can cause some problems in merging the two images. Another possible solution is to change Equation (22), taking into consideration the ambiguities. We then consider

$$\frac{de}{d\theta} = |sin(\theta)| \tag{25}$$

which leads to

$$e(\theta) = \begin{cases} 1 - \cos\theta & \text{if } \theta \geq 0 \\ \cos\theta - 1 & \text{if } \theta \leq 0 \end{cases} \tag{26}$$

Thanks to Equation (26), it is possible compute the spectral coverage in the $k_e$ domain, using the derivative chain rule:

$$k_e = \frac{4\pi}{\lambda} \left[ \frac{\partial R(\tau, \mathbf{p})}{\partial x} \frac{\partial x}{\partial \theta} \frac{\partial \theta}{\partial e} + \frac{\partial R(\tau, \mathbf{p})}{\partial y} \frac{\partial y}{\partial \theta} \frac{\partial \theta}{\partial e} \right] \tag{27}$$

Plugging Equation (25) in Equation (27) one gets

$$k_e(\tau) = \begin{cases} \dfrac{4\pi}{\lambda} \dfrac{rx(\tau)}{R} & \text{if } \theta \geq 0 \\ -\dfrac{4\pi}{\lambda} \dfrac{rx(\tau)}{R} & \text{if } \theta \leq 0 \end{cases} \tag{28}$$

The spectral descriptor has sudden change of sign around $\theta = 0$. This can become an issue for the space invariance property that we require from our custom reference system. In particular, due to the dependency on $\theta$ in Equation (28), the space spectral descriptor will lose it's space invariance property. For instance, let us assume to have a target at a positive angle $\theta$ and to have only positive values of the radar sensor $x$. The covered tile in the spatial spectral domain will be purely positive. On the other hand, if we have a target with a negative angle $\theta$, only negative wavenumbers will be covered, which is not the spatial invariance property we would like to have. Anyway, in order to make the spectral contributes of all the targets in the scene sum up in the same spectral portion, the synthetic aperture must be centered around $x(\tau) = 0$; in addition, an odd number of bursts must be processed. This must be done in order to exploit the odd symmetry of the spatial spectral estimator, in order to let the targets' spatial spectral component sum up in the same spectral portion.

### 4.3. Constant Spectrum Approximation and Validity Limits

In order to achieve a space spectral invariance property in Equation (28), we would like to consider as a first approximation

$$r \simeq R \tag{29}$$

therefore, we define a validity region for this approximation, where the value of $r$ is comparable with the total synthetic aperture length. In addition, in order to take into account the burst acquisition mode, we will introduce an indicator function $I(x(\tau))$, which takes the value 1 if the radar is acquiring the data and 0 otherwise.

By taking into account the above considerations, Equation (28) becomes

$$\bar{k}_e = \frac{4\pi}{\lambda} I(x(\tau)) x(\tau) \tag{30}$$

By using Equation (30), it is possible to compute the spatial spectral bandwidth and the central wavenumber in a straightforward manner. The spectral bandwidth can be computed by considering the displacement of the vehicle within one burst:

$$B = \frac{4\pi}{\lambda} \Delta x = \frac{4\pi}{\lambda} L_{synth} \tag{31}$$

where $\Delta x$ is the displacement along the direction of motion, and $L_{synth}$ refers to the total synthetic aperture. In addition, the previous equation can be used to state the resolution in the $(r, e)$ reference system, indeed:

$$\rho_e = \frac{2\pi}{B} = \frac{\lambda}{2L_{synth}} \tag{32}$$

where $\rho_e$ is the resolution along the $e$-variable. Similarly, it is possible to define the central wavenumber by using the midpoint of the synthetic aperture $x_0$, as

$$k_{e_0} = \frac{4\pi}{\lambda} x_0 \tag{33}$$

The above equations can be referred to a constant spectrum approximation (CSA). Actually the CSA presents some limits, in particular outside the validity region defined thanks to Equation (29), i.e., in the *near range zone*, where the range becomes comparable to the synthetic aperture length. However, thanks to the CSA, it is possible to predict exactly the image spectral component according to the two laws expressed in Equation (31) and in Equation (33).

### 4.4. Residual Motion Compensation and Phase Calibration

This subsection describes at high level the autofocus and calibration procedure performed on each single burst after their focusing. We remark that these procedures are mandatory to remove the residual trajectory error and to be able to jointly exploit all the bursts, achieving in this way extremely fine resolutions.

#### 4.4.1. Residual Motion Compensation

An important aspect of accurate SAR imaging is the knowledge of the platform trajectory, which should be known with an accuracy of a fraction of a wavelength [10]. A threshold can be set by simply requiring that the velocity error must be smaller than the velocity resolution of the system:

$$\Delta v_{ego} = \frac{\lambda}{2T_c} \tag{34}$$

where $T_c$ is the synthetic aperture time , i.e., the time taken by the vehicle to move along the synthetic aperture. An error larger than $\Delta v_{ego}$ will result in a mis-localization of the target of more than a resolution cell. Small wavelengths and high integration times call for a very high accuracy in the knowledge of the trajectory, which is typically an unmet condition in standard navigation units. This condition calls for a dedicated procedure to refine the platform motion directly from radar data. The residual motion compensation procedure used during this work has been fully developed and explained in [18]. In this paper, we will briefly recall the main concepts.

The availability of a multichannel device greatly simplifies the task of residual velocity estimation, which can be carried out to within sufficient accuracy by analysis of the Doppler frequency at different off-boresight angles, see for example [3]. The most common model in literature is the constant *velocity* error. This model is generally valid over short synthetic apertures [3,21–23]. This hypothesis leads to an easy expression of the residual Doppler frequency over each target in the scene:

$$f_d(\theta) = \frac{2}{\lambda}\Delta v_{ego,r}(\theta) = \frac{2}{\lambda}(\Delta v_{ego,x}\cos\theta + \Delta v_{ego,y}\sin\theta) \tag{35}$$

where $\Delta v_{ego,r}(\theta)$ is the residual radial velocity as seen by a target at angular position $\theta$, $\Delta v_{ego,x}$ is the residual velocity along $x$, while $\Delta v_{ego,y}$ is the one along $y$. The estimation of the parameters of interest ($\Delta v_{ego,x}$ and $\Delta v_{ego,y}$) can be carried out by detecting a set of $N$ fixed Ground Control Points (GCPs) in the scene and by retrieving the residual Doppler frequency over each of them. In this way we obtain a linear system of equations:

$$\begin{bmatrix} f_d(\theta_0) \\ f_d(\theta_1) \\ f_d(\theta_2) \\ \vdots \\ f_d(\theta_N) \end{bmatrix} = \frac{2}{\lambda}\begin{bmatrix} \cos\theta_0 & \sin\theta_0 \\ \cos\theta_1 & \sin\theta_1 \\ \cos\theta_2 & \sin\theta_2 \\ \vdots & \vdots \\ \cos\theta_N & \sin\theta_N \end{bmatrix}\begin{bmatrix} \Delta v_{ego,x} \\ \Delta v_{ego,y} \end{bmatrix} \tag{36}$$

Notice that the measurement vector containing all the residual Doppler frequencies can be obtained by a simple Fourier Transform of the slow-time over each detected GCP in the scene. This is the case thanks to the assumption of constant velocity error that in turn is responsible for a constant residual Doppler frequency. The design matrix is built upon the knowledge of the off-boresight position of each GCP. In this setting, a multichannel device is essential to provide an a priori knowledge of the angular positions of the GCPs. The system of Equation (36) can be inverted using a simple least squares (LS) approach obtaining an estimate of the residual velocities.

### 4.4.2. Phase Calibration

Once the residual motion compensation procedure has been completed, each burst is properly focused. A further calibration step is now necessary to compensate for the displacement between single bursts apertures. Let us assume that we have a set of $N_{burst}$ SAR images and that they are not aligned with each other. The displacement can be modeled as a phase shift between bursts as

$$\phi(\mathbf{x}) \simeq -\mathbf{k}(\mathbf{x})^T \Delta x \tag{37}$$

where $\mathbf{k}(\mathbf{x})$ is the wavevector associated with the spatial position $\mathbf{x}$ and $\Delta x$ is the residual displacement between two bursts. In order to explain the phase calibration procedure, let us consider $N_{burst} = 2$, i.e., a couple of SAR images, one as the master image and the other as the slave one. Let us also consider a set of GCP shared between the two; more precisely, we will refer to $a_i^M$ as the complex amplitude of the $i$th GCP of the master image, and to $a_i^S$ as the complex amplitude of the $i$th GCP of the slave one. The procedure consists in maximizing the following expression w.r.t. $\Delta x$:

$$\max_{\Delta x}\left|\sum_i^{N_{GCP}} a_i^M + a_i^S \times \exp\{j(\mathbf{k}_i(\mathbf{x}_i)^T\Delta x)\}\right| \tag{38}$$

where $\mathbf{k}_i(\mathbf{x}_i)$ is the wavevector associated with the spatial position $\mathbf{x}_i$ of the $i$th GCP.

In practice, this procedure consists in an exhaustive search that takes place for each dimension of the images. Once the optimum value of $\Delta x$ has been found, a phase field is

applied to the slave image in order to compensate for the residual phase shift. In the end, the final image is computed as the sum of the two images.

### 4.5. The CLEAN Algorithm

The CLEAN algorithm was first introduced for removing grating lobe induced artifacts in astronomy [7] and also is widely adopted in radar imaging [11–14]. It consists in successively selecting bright targets and removing their side-lobe responses by subtracting the impulse response of each target individually in an iterative manner. Our use-case lies in a class of techniques called compressive sensing, which aim at reconstructing a signal, given an under-determined system to be solved; those problems are commonly used to tackle unevenly sampled, periodically recurring, gapped, arbitrarily sparse data problems, which are common in SAR data processing and in astronomy [7,8,11–14,24,25]. We chose CLEAN as a target estimation algorithm for our work due to its low computational cost. Because of this, CLEAN is suitable for a real time implementation as required by the automotive application. This algorithm works under the assumption of impulsive sources to be estimated. Generally speaking, given a generic discrete signal $s(n)$, CLEAN performs a peak estimation, i.e., it selects the maximum value of $s(n)$ as the signal source and then it gives as result a new signal $\tilde{s}(n)$, made up by the highest peaks of $s(n)$.

More accurately, let us define a generic discrete signal $s(n)$, of $N_t$ elements and its sparse spectrum $S(\omega_n)$ of $N_\omega$ element, and assume that there exists a impulse response filter $h(n)$ known *a priori* such that it perfectly describes the unitary impulse response of the system. The signal $s(n)$ can be described as the sum of scaled impulses, namely:

$$s(n) = \sum_{k=1}^{N_t} \alpha_k \delta(n - k) \tag{39}$$

where $\alpha_k$ is a complex value describing the amplitude of the signal. The CLEAN algorithm performs a peak estimation by taking the position of the maximum of $s(n)$ and its value, as

$$k = \arg\max_n s(n) \tag{40}$$

$$\alpha = s(k) \tag{41}$$

where $k$ is the position of the maximum and $\alpha$ its value. A new impulsive discrete signal is now formed by considering a single peak in the position of the maximum ($k$) with a value defined as before:

$$\hat{s}(n) = \alpha \delta(n - k) \tag{42}$$

Then, the impulse response of the estimated source $\bar{s}(n)$ is computed thanks to a convolution as

$$\bar{s}(n) = \sum_{k=0}^{n} \hat{s}(k) h(n - k) \tag{43}$$

where $h(n)$ is the filter describing the unitary system impulse response. The next step of the CLEAN algorithm involves the computation of an error sequence $e(n)$, as

$$e(n) = s(n) - \bar{s}(n) \tag{44}$$

After that, the CLEAN procedure is iteratively applied to $e(n)$, until stopping criteria are met. In our work, we have used a threshold under which the signal is considered pure noise and a maximum number of peaks are to be found; as a rule of thumb, 20 targets were selected. The advantage of CLEAN is that it both performs the side-lobes removing procedure and a denoising procedure, since as said before, $\tilde{s}(n)$ is indeed an impulsive signal made up by the estimated peaks of $s(n)$.

## 5. Results

In this section, the results of our work are shown. The data have been acquired during an acquisition campaign performed with an eight-channel radar called ScanBrick $^{\circledR}$ developed by Aresys $^{\circledR}$ and working at 77 GHz. The ScanBrick $^{\circledR}$ used a bandwidth $B = 1$ GHz, which provides a range resolution of about 15 cm. In Table 1, the system parameters are summed up. For a more detailed explanation of the hardware set up of the experiment, see [16].

**Table 1.** Data Parameters

| Parameters | Value |
| --- | --- |
| Carrier frequency $f_c$ | 77 GHz |
| Bandwidth $B$ | 1 MHz |
| Active Tx channels | 2 |
| Active Rx channels | 4 |
| Geometry (mode) | forward looking |
| Range Resolution | 15 cm |
| Synthetic aperture length | 1.66 m |
| Angular Resolution | $6.58 \times 10^{-2}$ deg |

The Radar data are accompanied by the IMU data in order to know the location of the vehicle with respect to a Cartesian external reference system. The vehicle is also equipped with a camera mounted on the car rooftop in order to provide the optical ground truth of the scene.

The radar is mounted in a forward looking configuration; therefore, the radar boresight is pointing in the direction of motion. The eight-channel radar is implemented with two transmitting antennas $N_{Tx} = 2$ and with four receiving ones $N_{rx} = 4$, leading to eight virtual monostatic antennas spaced by $\lambda/4$. The angular resolution provided by an eight-channels radar is about 14 deg. Each burst is a portion of the trajectory made of $M = 256$ slow time samples. In each burst, the car travels for as little as 30 cm which would lead to a maximum resolution of 0.37 deg (at the right or the left of the car).

On the other hand, the whole aperture comprehensive of both on and off times, as in Figure 4, is about 1.66 m, which leads to an extremely fine angular resolution of $6.58 \times 10^{-2}$ deg: five times finer than the angular resolution provided by one burst.

The radar uses a multiplexing scheme known as time division multiplexing (TDM). It consists in delaying the signal transmission from the second Tx-antenna in order to not create interference. The delay is given by the PRI. For this reason, there is a spatial shift between the two virtual antenna clusters. Indeed, since the vehicle is moving at a given velocity and the transmission is delayed, the second antenna cluster is shifted in space. This shift can be computed since both the velocity and the PRI are known by the system. This was not an issue, since the TDBP algoritm we used for focusing the images takes into consideration the position of the antennas phase centers and compensates, exactly, the antenna shift. Thus, no artifacts due to the TDM arose.

The SAR image after the phase calibration procedure, considering the whole aperture, is depicted in Figure 7a. The image is dense of peaks as an effect of the aberrations due to the burst acquisition mode. After the grating lobes suppressing procedure using $N_{max} = 20$ as the maximum number of targets per-row, the image shown in Figure 7b has preserved the brightest targets. Let us notice that, thanks to the CLEAN algorithm, a point target reaches one-pixel resolution.

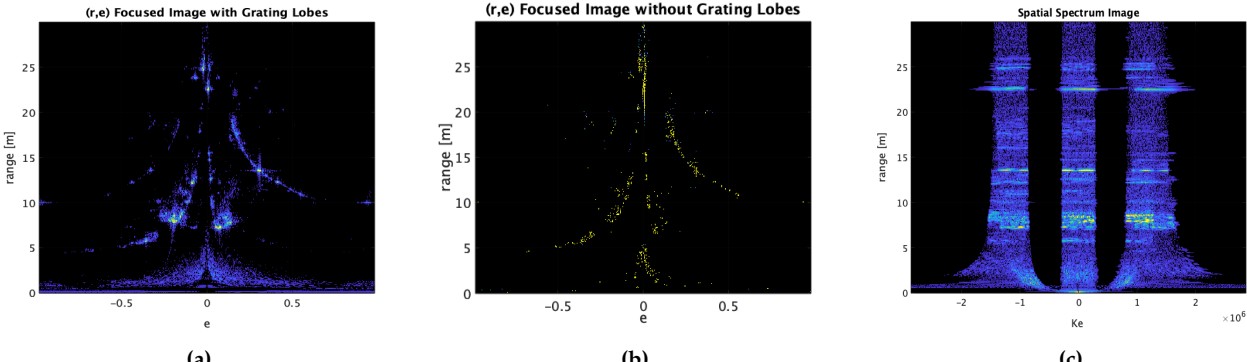

**Figure 7.** (**a**): Scene focused in the (r,e) reference system. (**b**): CLEAN processed image, the image is much less dense since the grating lobes have been removed. (**c**): Spatial spectrum image, for each burst there is a column of data which is space invariant in the validity region of the CSA.

The spectrum of the grating lobes corrupted image is depicted in Figure 7c. The spatial spectrum is consistent with the theory developed in the previous sections. The spectral estimation, as said before, clearly shows a validity zone where the gaps are observable. The size of each tile is proportional to the length of the synthetic aperture exploited by each burst, as expected from Equation (30). In the near range, instead, the CSA is no longer valid and we see that the three tiles in the spectrum merge around $r = 0$.

Both the Figure 7a and Figure 7b have been projected into the Cartesian domain, respectively, in Figure 8a,b. In the former, it is difficult to distinguish the shapes of the objects due to the grating lobes presence and details not being sharp; in the latter, instead, the details are sharper and the resolution is increased. A rough comparison can be performed with the optical image in order to identify the shapes of the objects in the scene. The optical image is presented in Figure 9.

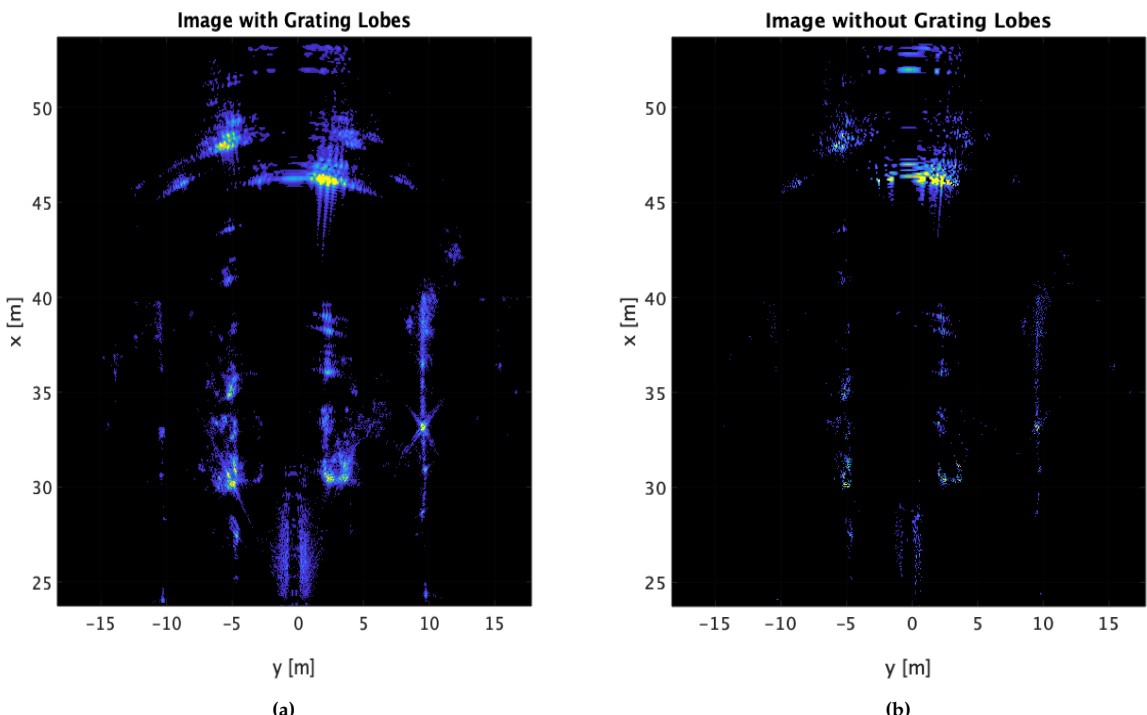

**Figure 8.** (**a**): Image geocoded into the Cartesian reference system. (**b**): Image geocoded into the Cartesian reference system without the grating lobes.

We would like to draw attention to two particular objects in the images. The gate in Figure 10c has been zoomed in the SAR images and can be found in Figure 10a,b. In the former, the grating lobes can be seen in a clear manner since the effect is very similar to a blurring effect. In Figure 10b, instead, the bright back-scattering points have a shape that is sharper.

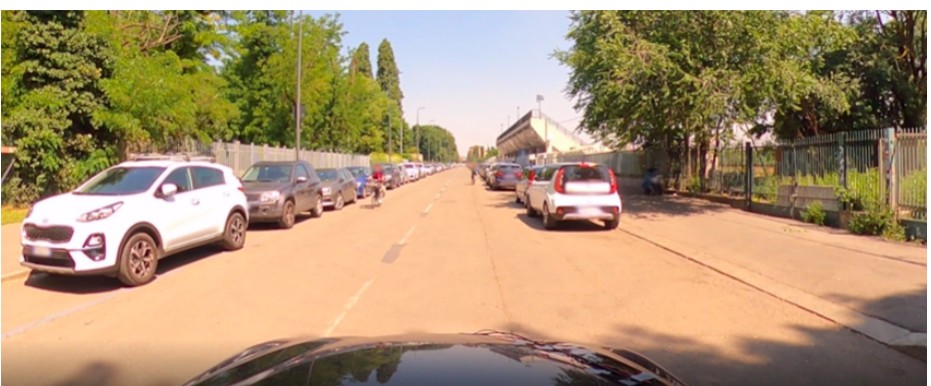

**Figure 9.** Optical reference taken from a camera placed on top of the vehicle.

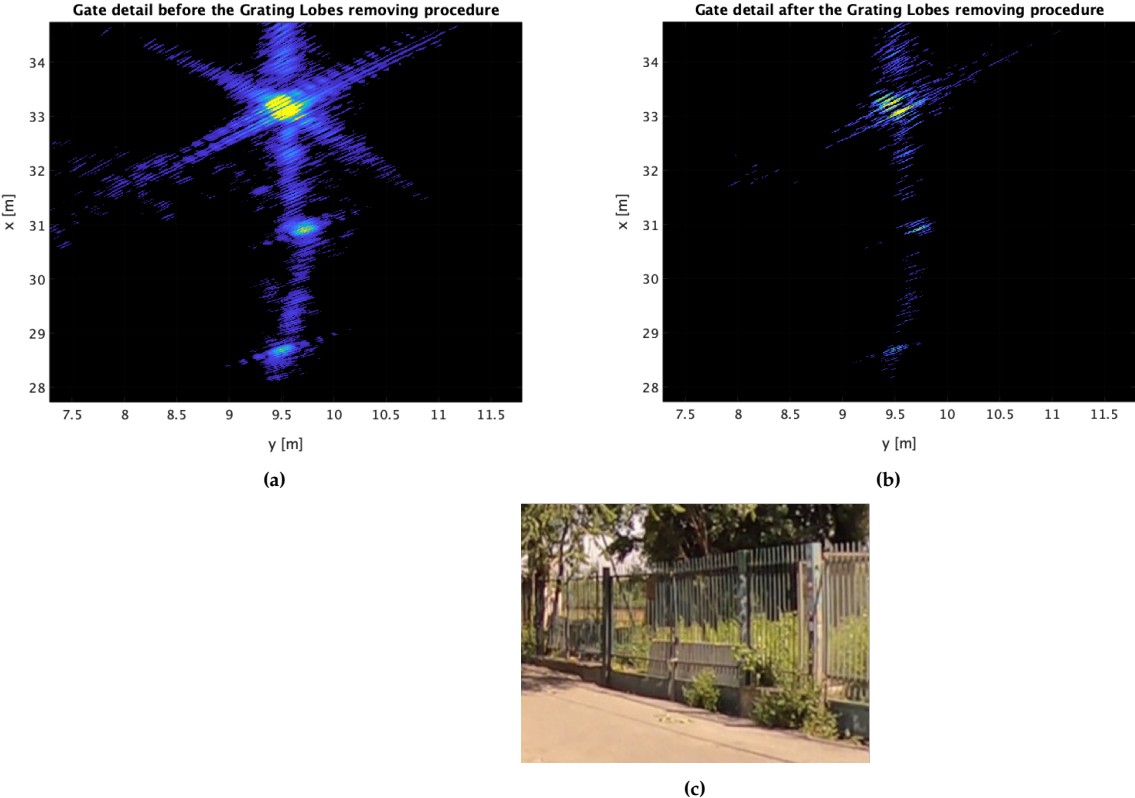

**Figure 10.** (**a**): SAR image of a gate detail before the grating lobes suppressing algorithm. (**b**): SAR image of a gate detail after the grating lobes suppressing algorithm. (**c**): Optical reference of the gate detail.

The other notable detail is depicted in Figure 11c, whose SAR equivalents are Figure 11a,b. In the latter, the shape of the parked car is much more clear, since details are recovered and grating lobes are suppressed.

In Figure 12, we propose a comparison between different processing with one or multiple bursts. Figure 12a represents the focused car by using a single burst. The resolution is rather poor, but at least no grating lobes are visible. Figure 12b shows the incoherent sum of the three bursts. In this case, noise is abated by the incoherent average, but there is

no significant resolution improvement. An even better imaging capability can be found in Figure 12c which is the coherent sum of the three SAR images. Some details start to emerge, but, on the other hand, the grating lobes artifacts are clearly visible. The last image in Figure 12d shows our processing; details are reconstructed and the shape of the car is much more defined. Sidelobes from strong reflecting surfaces are also suppressed (see the bottom left of the image).

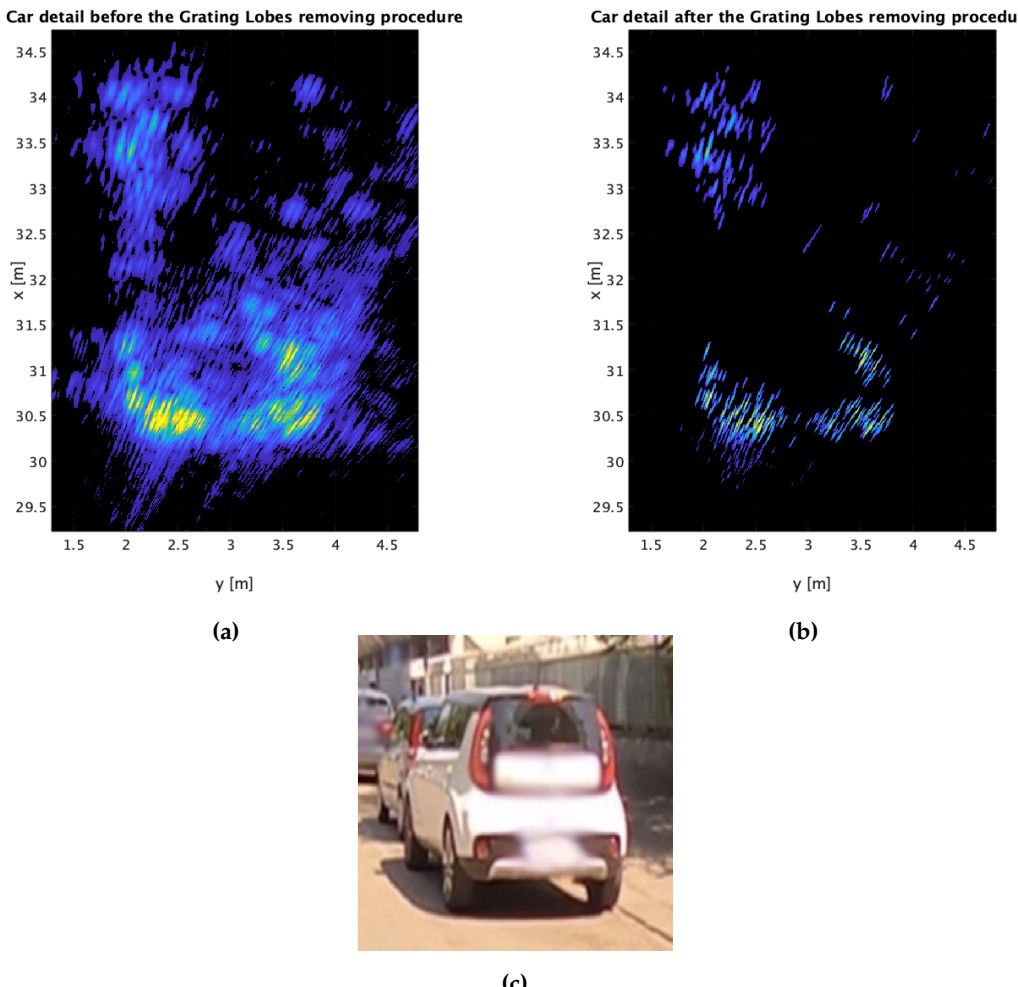

**(a)**

**(b)**

**(c)**

**Figure 11.** (**a**): SAR image of a car detail before the grating lobes suppressing algorithm. (**b**): SAR image of a car detail after the grating lobes suppressing procedure. (**c**): Optical reference of the car detail.

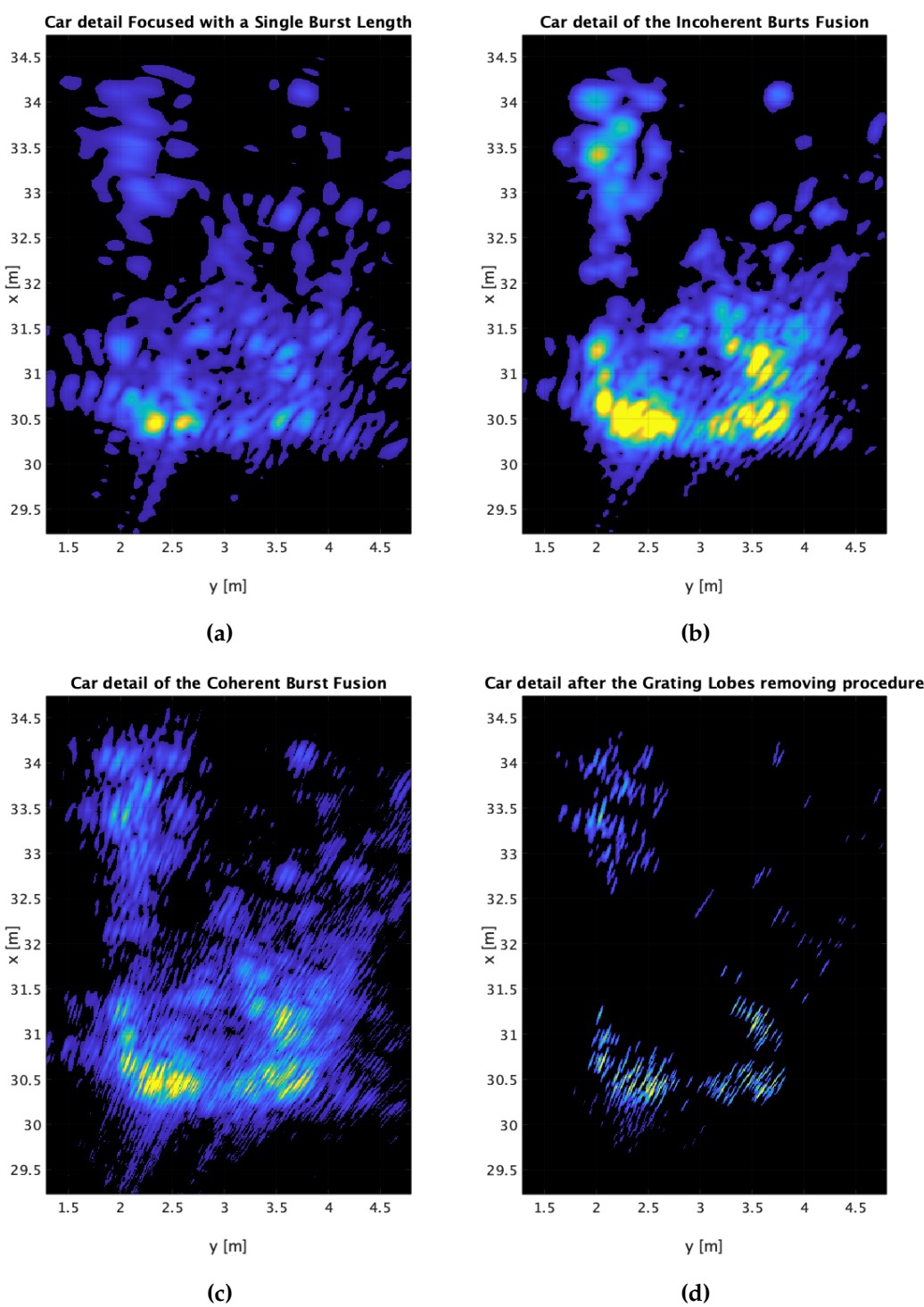

**Figure 12.** (**a**): Car detail of a SAR image generated using the synthetic aperture of a single burst length. (**b**): Car detail of a SAR image generated using the incoherent sum of the three bursts. (**c**): Car detail of a SAR image generated using the coherent sum of the three bursts. (**d**): Car detail of a SAR image generated using the grating lobes removing procedure.

The previous comments can be written also for Figure 13. In Figure 13d, the single poles can be spotted in the images as point scattering targets after the grating lobes removing procedure. The information retrieved by the obtained images can be merged with that coming from cameras and LiDAR for a future step ahead into the autonomous driving.

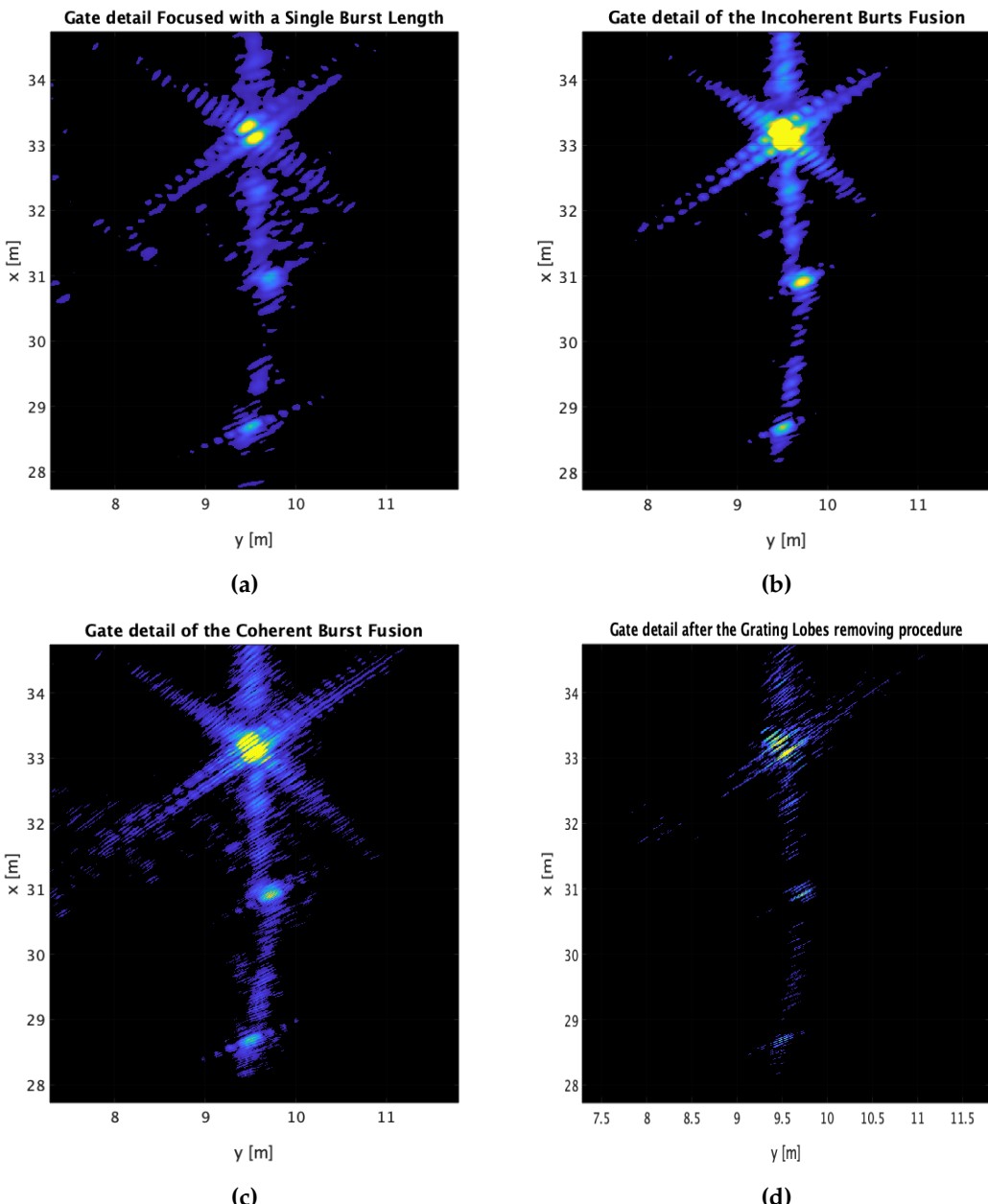

**Figure 13.** (**a**): Gate detail of a SAR image generated with the synthetic aperture of a single burst length. (**b**): Gate detail of a SAR image generated using the incoherent sum of the three bursts. (**c**): Gate detail of a SAR image generated using the coherent sum of the three bursts. (**d**): Gate detail of a SAR image generated using the grating lobes removing procedure.

## 6. Conclusions

In this paper, we have presented an algorithm to remove grating lobes from vehicle-based SAR images. Automotive radars are required by the law to operate in a non-continuous manner. While this condition does not represent an issue in classical automotive radar imaging, this is not the case for automotive SAR. Indeed, SAR processing is performed by exploiting a set of pulses acquired continuously from different spatial positions. If the transmission is not continuous, however, there are some gaps in the illuminated spatial spectrum of the scene; therefore, the reconstructed image will contain some aberrations called *grating lobes*.

This paper proposes a complete mathematical description of the spatial spectrum of a SAR image focused using burst data. A workflow to correct grating lobes is then

presented. We start by detailing why a Cartesian or polar reference system cannot be used to solve this problem. We then continue by deriving a suitable reference system, which is a transformation of the polar one allowing for a space-invariant spectral representation. In other words, the spectrum of the image will depend only upon the position of the sensor and not of the target itself. The workflow continues by exploiting a compressive sensing algorithm called CLEAN, which allows the recovery of an extremely fine-resolution image without grating lobes.

The whole process has been validated through a real data set acquired with an automotive radar operating in W-band, mounted on a vehicle in forward-looking configuration. The procedure proposed in this paper has proven to remove the aberration in the image, leading to a clear recognition of the targets and their details.

**Author Contributions:** Conceptualization, M.G.P., M.M. and S.T.; methodology, S.T.; software, M.G.P.; validation, M.G.P. and M.M.; formal analysis, M.G.P. and S.T.; investigation, M.G.P., S.T., M.M., A.M.-G., C.M.P. and I.R.; resources, S.T., A.M.-G. and C.M.P.; data curation, M.G.P.; writing—original draft preparation, M.G.P. and M.M.; writing—review and editing, All; visualization, M.G.P. and M.M.; supervision, S.T. and C.M.P.; project administration, S.T. and I.R.; funding acquisition, S.T. and I.R. All authors have read and agreed to the published version of the manuscript.

**Funding:** This research was funded within under the program of the Joint Research Lab with participants Politecnico di Milano and Huawei Technologies Italia.

**Institutional Review Board Statement:** Not applicable.

**Informed Consent Statement:** Not applicable.

**Data Availability Statement:** Not applicable

**Acknowledgments:** We would like to thank Paolo Falcone from Aresys for the great support provided with all hardware-related matters during acquisition campaigns.

**Conflicts of Interest:** Dr. Ivan Russo is currently a Huawei Technologies employee. The company approved the publication of the paper with him as a co-author.

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
