# Peer review of "Very High Resolution Automotive SAR Imaging from Burst Data"

_remotesensing, doi:10.3390/rs15030845_

Round 1
Reviewer 1 Report (Previous Reviewer 2)
The authors made sufficient changes to the manuscript to address all comments from my first review. Thank you!
Reviewer 2 Report (Previous Reviewer 1)
Thank you for your answers. The integral to which I refered was in Equation (23) (1st version), resp. (24) (2nd version). Check spelling in your corrections, i.e.,:
- Page 4: N_{ch}
- Page 5: Figure 3
I have no further comments and recommend publication.
This manuscript is a resubmission of an earlier submission. The following is a list of the peer review reports and author responses from that submission.
Round 1
Reviewer 1 Report
The authors introduce a new approach to remove grating lobes in automotive SAR imaging. To this end they describe SAR imaging for apertures with gaps as they arise in automotive and a post-processing method to remove the corresponding grating lobes. The principles are well-explained and the methods and the results are interesting. The overall presentation is clear, however I have a few comments/ questions:
- What are the practical impacts of the results of the article for automotive SAR? The author mention simply in the introduction that grating lobes could lead to false detections and give two rather old references. E.g. in Figure 13(d) one observes that the grating lobes disappear but also the scattering centre seems to be less visible.
- At which point is the (r,e) reference system used in the algorithms? It seems there is a gap in the spectrum. How do you "fill" the gap? Why don't you work with the gap in spatial domain?
- Do you need requirements on the velocity of the car to fullfil the spatial sampling theorem?
- It is said that the CLEAN-algorithm uses 20 peaks. Does it mean that only 20 non-zero elements should be visible in the image 7(b)?
- Image 7(b) is hardly visible.
Further Comments:
- Line 32: It is written that the usage depends on the wavelength, however it is said that a frequency of 77GHz is used.
- After line 148: What is \theta_T?
- Line 128: Maybe it would be better to use an indefinite integral, which gives an arbitrary integration constant, also with respect to Equation 25.
- There seems to be no reference to Figure 3 in the text.
- In Line after 180: What is the synthetic aperture time?
Typos:
- Line 4: "do not allows ..."
- Line 75/76: "Notice that now ..."
- Line 92/93: "This represantation comes ..."
- Line 95: I think \nabla sould be gradient operator, not the Laplacian operator
- Line 124: Write "we" in capital letter
- Line 127: "For a artificial problem ..."
- Line 139: "for a give"
- After line 151: Put a space after however in: "However,this solution can"
- After Line 168: Reference is missing: "??"
- After Line 168: "and the central wavenumber is a straight ..."
- Line 254: "Grating lobes" in lowercase letters.
Reviewer 2 Report
An interesting and very well-written paper describing the implementation of automotive SAR using spatially sparsely sampled data and the suppression of the resulting grating lobes.Comments: - Line 72: Why is the spacing between the elements lambda/4 and not lambda/2 which is commonly used to achieve a non-ambiguous -90° … +90° angular range? - I wonder if CLEAN is an appropriate method to use here. In general, CLEAN is good when there is a very high dynamic range between the targets in the resolution cell, i.e. one of the targets is much stronger than the other. In this case the estimate of the strong target is very good and it can be effectively subtracted. In your sparse SAR application, the dynamic range is low, i.e. the targets and the grating lobes all have similar strength and the estimate of the strongest target’s parameters will very likely be biased by the superposition with the grating lobes of other targets. In this case the subtraction will not be effective leading to artifacts and likely false positives. Can you please comment on this and give positive and negative examples of where CLEAN actually helped and where it caused a problem. Maybe the algorithm RELAX would work better here. - Comparing Figure 9 (a) and (b) it looks like the sidelobe-removal method created a very strong target at around [-2 m, 47 m] in (b) which isn’t there in (a). This is also the case at around [0 m, 52 m]. Can you please comment on this? - I am having a hard time finding examples where CLEAN actually helped. The figures 9-12 with removed grating lobes mostly look like thresholded versions of the figures with the grating lobes. Can you please show a few concrete examples where CLEAN actually leads to a performance improvement and compare it to simple thresholding? - The used radar system is described in section “5 Results”. Maybe it will be clearer if it is described in a separate section / subsection “Radar Hardware”. - You mention that the radar system is a 2-Tx MIMO system. Can you please also give some information about the multiplexing scheme used (TDM, FDM, DDM, CDM, …) and if that caused any additional challenges / artifacts.
Minor problems: - Line 64: “and and” -> “and” - Line 75: “the is” -> “there is” - Line 123: “gaps is the spectrum” -> “gaps in the spectrum” - Line 124” “we recall” -> “We recall” - Between (19) and (20): “lets’” -> “let’s” - Line 165: move leading comma to (28) and “the the” -> “the” - Line 168: “Eq.??” - Line 177: “exploits” -> “exploit” - Line 203: “ad” -> “and” - Line 224: “15cm” -> “15 cm” and don’t italicize cm